# Wedge Resection and Optimal Solutions for Invasive Pulmonary Fungal Infection and Long COVID Syndrome—A Case Report and Brief Literature Review

**DOI:** 10.3390/reports7020025

**Published:** 2024-04-05

**Authors:** Ioana-Madalina Mosteanu, Beatrice Mahler, Oana-Andreea Parliteanu, Alexandru Stoichita, Radu-Serban Matache, Angela-Stefania Marghescu, Petruta-Violeta Filip, Eugen Mota, Mihaela Ionela Vladu, Maria Mota

**Affiliations:** 1“Marius Nasta” Institute of Pneumophtiziology, 050159 Bucharest, Romania; madalina.mosteanu@yahoo.com (I.-M.M.); oana_andreea@yahoo.com (O.-A.P.); alexandru.stoichita@drd.umfcd.ro (A.S.); 2Doctoral School, University of Medicine and Pharmacy of Craiova, 200349 Craiova, Romania; eugenmota@yahoo.com (E.M.); mihmitzu@yahoo.com (M.I.V.); mmota53@yahoo.com (M.M.); 3Faculty of Medicine, “Carol Davila” University of Medicine and Pharmacy, 050474 Bucharest, Romania; radu.matache@gmail.com (R.-S.M.); tefania_angela@yahoo.com (A.-S.M.); petruta.filip@umfcd.ro (P.-V.F.); 4Department of Thoracic Surgery, “Marius Nasta” Institute of Pneumophtiziology, 050159 Bucharest, Romania; 5Department of Research, “Marius Nasta” Institute of Pneumophysiology, 050159 Bucharest, Romania; 6Department of Gastroenterology and Internal Medicine, Clinical Emergency University Hospital, 050098 Bucharest, Romania; 7Department of Diabetes, Nutrition and Metabolic Diseases, County Clinical Emergency Hospital of Craiova, 200642 Craiova, Romania

**Keywords:** invasive pulmonary aspergillosis, long COVID-19, diabetes, chest imaging, candidiasis, immunosuppression

## Abstract

A rise in fungal infections has been observed worldwide among patients with extended hospital stays because of the severe infection caused by the new coronavirus pandemic. A 62-year-old female patient was admitted with a severe form of Coronavirus disease 2019 (COVID-19) and spent four weeks in the intensive care unit (ICU) requiring mechanical ventilation support before being moved to a tertiary hospital for further testing. *Aspergillus fumigatus* filamentous fungus, *Candida* spp., and positive bacteriology for multidrug-resistant Klebsiella pneumoniae and Proteus mirabilis were identified by bronchial aspirate cultures. The patient’s progress was gradually encouraging while receiving oral antifungal and broad-spectrum antibiotic therapy along with respiratory physical therapy; but ultimately, thoracic surgery was necessary. Long-lasting tissue damage and severe, persistent inflammatory syndrome were the two main pathophysiological mechanisms that led to significant outcomes regarding lung lesions that were rapidly colonized by fungi and resistant flora, cardiac damage with sinus tachycardia at the slightest effort, and chronic inflammatory syndrome, which was characterized by marked asthenia, myalgias, and exercise intolerance.

## 1. Introduction

The severe acute respiratory syndrome coronavirus-2 (SARS-CoV-2) pandemic has spread at an unprecedented rate throughout the world since it was first discovered in Wuhan, China, in December 2019 [1]. Immunosuppression and its accompanying consequences were widespread because of the cytokine storm triggered by the new virus, along with the host’s risk factors, and the necessary medication given to these patients throughout their hospital stay.

Viral infection results in dysregulation of the immune system, either locally or systemically, and reduces ciliary clearance [2]. Invasive pulmonary aspergillosis (IPA), a dangerous condition that affects immunocompromised individuals, is caused by Aspergillus hyphae invading the pulmonary vasculature. The diagnosis and treatment of invasive pulmonary aspergillosis have advanced to encompass emerging risk factors, such as critically ill people, individuals with chronic obstructive lung disease, and individuals with liver illness [3,4,5].

Furthermore, a lack of previous immunization through vaccination or disease set the stage for the serious development of an omicron variant infection, according to recent findings supporting that vaccination decreased the likelihood of being hospitalized with omicron [6,7].

Like many other viral diseases, a variety of chronic symptoms have been reported following the acute infection. Although post-COVID-19 symptoms or “Long COVID” are frequently discussed, there is still more evidence needed to support them. Breathlessness, cognitive decline, exhaustion, anxiety, and sadness are the major characteristics of a tentative description of persistent symptoms and probable sequelae that last longer than four weeks after the onset of the condition [8]. 

## 2. Detailed Case Description

### 2.1. Clinical Findings

A 62-year-old female patient, known to have high blood pressure and type 2 diabetes who had undergone treatment with Metformin for 5 years and was recently diagnosed with a severe form of COVID-19, was transferred from an emergency hospital to the “Marius Nasta” Institute of Pulmonology for further investigations and specialized diagnosis. According to the medical data from the previous hospital, the patient was admitted to the ICU for several weeks due to severe COVID-19 infection and needed invasive mechanical ventilation. The patient worked as a welder, she was a nonsmoker, and she was unvaccinated against SARS-CoV-2. At admission the patient had a poor general condition, she was overweight (Body Mass Index 26,14 kg/m^2^) and self-mobilization was not possible, as she presented significant desaturation on exertion (96% maintained by an oxygen flow of 3 L/min through the nasal cannula) and sinus tachycardia since she was hospitalized for over a month in the previous hospital. 

### 2.2. Blood Workup, Functional Tests, Bronchoscopy, and Radiological Findings

The blood tests showed hyperglycemia, mild hepatocytolisis syndrome, mild hypopotassemia, mild normocytic normochromic anemia, and inflammatory syndrome with the values shown in Table 1.

Functionally, the patient had a mild restrictive ventilatory dysfunction but with a severely low diffusing capacity for carbon monoxide (TLCO) (Table 2). 

The chest computer tomography (CT) examination revealed pulmonary consolidations with a central cavity with a heterogeneous air content and polypoid tissue areas at the level of the right upper lobe, right lower lobe, and left lower lobe. These lesions were accompanied by ground glass infiltration, drainage bronchi, adjacent bronchiectasis, and mediastinal lymphadenopathy images and the overall imagistic diagnosis indicated invasive pulmonary aspergillosis (Figure 1).

The bacteriological and fungal examination of the sputum showed the presence of Proteus mirabilis and over 100 colonies of filamentous fungi.

The bacteriological examination identified Klebsiella pneumoniae-XDR (extensively drug-resistant) and Proteus mirabilis in the bronchial aspirate. Also, for the fungal examination, more than 100 colonies of filamentous fungi of *Aspergillus fumigatus* and colonies of *Candida* spp. were discovered.

### 2.3. Treatment and Monitoring

The patient received intravenous antibiotics with glycylcyclines and aminoglycosides in aerosols, an oral antifungal (mentioning that loading dose was administered in the previous hospital as in 800 mg in the first day), an injectable anticoagulant treatment, oral mucolytics, and anticholinergic bronchodilator aerosols, as well as inhaler devices, beta-blockers, and HCN (hyperpolarization-activated cyclic nucleotide-gated) channel blockers, short-acting insulin as needed depending on glycemic values, hydro electrolytic rebalancing, and intestinal protection associated with the treatment (Table 3).

The respiratory physiotherapy program consisted of walking retraining exercises, toning and strengthening the muscles of the lower and upper limbs, diaphragmatic breathing, and recovery for the lower limbs using a bicycle.

Three serial chest X-rays were performed 3 days apart and showed a dynamic reduction in the size of the described opacities (Figure 2).

After a 2-week hospitalization, the patient was discharged with the ability to mobilize and feed herself, with an improvement in her general condition, but oxygen was still necessary at a flow of 2 L/min. The recommendations at discharge were to continue oral antifungal treatment with voriconazole for 3 months.

### 2.4. Follow-Ups

The follow-ups were carried out over a year with medical evaluation at one month, 3 months, 7 months, and 1 year, respectively. The investigations performed are summarized in Table 4. At each medical visit, the patient underwent a chest CT examination to see the imaging evolution of the pulmonary lesions.

In the first month of follow-up, the patient still presented sinus tachycardia (130 bpm) and she no longer needed oxygen therapy, as her oxygen saturation was 97% in atmospheric air. The chest CT exam showed the persistence of the lesions described during the hospitalization, but with the following reductions in their size: posterior right upper lobe 6.4/2.8 cm, apical right lower lobe 2/1.5 cm, lateral left lower lobe 3.6/3.4 cm (Figure 3).

After 3 months, the chest CT examination showed a favorable evolution (Figure 4), but further treatment with voriconazole was still recommended.

The patient followed a total of 7 months of treatment with voriconazole, until the chest CT examination found no significant improvement (Figure 5), and so she was referred to the thoracic surgery department. Since an economical resection of the lung parenchyma could be performed, a wedge resection was chosen, ensuring a better respiratory function for the patient in the postoperative period.

### 2.5. Pathology Report

The pathology report confirms the presence of pulmonary aspergillosis (Figure 6), which can be described as a lung fragment of 7 cm/3 cm, partially plated at the pleura with crepitations and reduced elasticity, presenting a central cavity area of 2 cm with necrotic-coarse detritus, burns, and an intracavitary surrounded by consolidated parenchyma.

The microscopic report shows necro”Ic ’asses containing Aspergillus microcolonies; in its thickness, it contained important granulation tissue, regenerative, reactive granulomatous giant-epithelioid and lesional aspects of bronchogenic granulomatosis. The perilesional parenchyma had foci of organized pneumonia (Figure 7).

Almost a year after admission the patient came for a follow-up and a chest X-ray was performed which showed a significant improvement in the radiological appearance (Figure 8). 

## 3. Discussion

We analyzed articles published between the years 2020 and 2023 about the association between COVID-19 and invasive pulmonary aspergillosis as well as the risk factors that contribute to a severe form of SARS-CoV-2 infection, using the PubMed search platform. The search words were “invasive pulmonary aspergillosis”, “long COVID” and “wedge resection” with 0 results on the search. After that, we widened the search using only “invasive pulmonary aspergillosis” and “long COVID”, with thirteen results, out of which five suited our case presentation best (Table 5). We searched for clinical cases of invasive pulmonary aspergillosis associated with SARS-CoV-2 infection to obtain a comparative analysis for the clinical case presented by us. The selection criteria for these cases were surviving female patients with diabetes and hypertension who had a severe form of COVID-19, were hospitalized in intensive care units, benefited from invasive mechanical ventilation, and during hospitalization were complicated with invasive pulmonary aspergillosis.

The diagnostic limitations in this case are represented by the following: (1) Our Institute is a monodisciplinary hospital, and the diversity of biological investigations is limited. To check if the patient’s diabetes was uncontrolled, it would have been useful to measure their glycosylated hemoglobin, but this was not possible because the laboratory does not have such tests; (2) additionally, serum indicators such as beta-D-glucan tests and galactomannan would have been useful, as the latter has a high specificity for invasive aspergillosis [14]. Galactomannan can also be detected in bronchoalveolar lavage sputum samples, and when a tuberculosis test is negative, a positive Aspergillus IgG test can be useful in identifying the chronic type of aspergillosis [15]. 

Also, the patient having more than 4 weeks of hospitalization with persisting clinical symptoms [16] allowed us to take into consideration that we are dealing with a long-COVID syndrome. This diagnostic assumption is supported by the patient’s long-term symptoms, including post-exertional malaise and shortness of breath over 13 weeks after COVID-19 infection (at the 6 MWT) along with heart palpitations with postural orthostatic tachycardia (considered a manifestation of the long-COVID syndrome).

Although most surgical operations are lobectomies, in particular cases it is always preferable to save parenchyma and benefit from VATS (Video Assisted Thoracic Surgery). The chosen surgical method was a wedge resection and this did not lead to a decrease in the functional capacity of the lung evidenced by spirometry, alveoli-capillary diffusion, and gas transfer factor methods. Lung parenchyma sparing is possible with early diagnosis, according to a study carried out in a Moroccan Center [17], in which half of the study group (51.48%) benefited from wedge resection, and therefore more wedge resections were performed on simple aspergilloma patients than on complex aspergilloma patients, with a statistically significant difference (*p* = 0.001). On the other hand, six individuals with PA (pulmonary aspergillosis) benefited from wedge resection in a study performed on the Chinese population [18], of whom one patient experienced a postoperative fungal rebound following a wedge resection by VATS. According to this study, opting for a full lobectomy can help prevent further complications and fungal relapses.

Patients who develop respiratory insufficiency necessitating intensive care and who have clinical symptoms consistent with COVID-19 and a positive RT-PCR (real-time–protein chain reaction) test should be specifically recognized to be at high risk for CAPA (COVID-19-associated pulmonary aspergillosis, because of the severe immune-system alteration intermediated by lymphopenia [19,20].

Wang et al. [21] observed early aggressive pneumoconiosis imaging alterations in patients with COVID-19 and IPA. Thus, in a retrospective review chart observed by Koehler et al. [22] the radiological damage during CAPA may also be present as combined ground-glass opacities with paving and peripheral nodular or interstitial consolidations, small nodular infiltrations, cystic cavities, and some air crescent signs.

Support for the use of GCs (glucocorticoids) in patients with critical COVID-19 who manifested with ARDS or other conditions that required invasive mechanical ventilation has been expressed by the World Health Organization [23,24]. Even so, because of the intricate quantitative and qualitative immunological dysregulations they cause [25], corticosteroids have been shown to increase a person’s vulnerability to invasive fungal diseases. It has previously been discovered that high-dose corticosteroids are linked to CAPA [26]. As a result, it is anticipated that the frequency of IFI may rise with the increased use of corticosteroids or other immunomodulating treatments [27]. Central venous catheters and the use of broad-spectrum antibiotics, otherwise commonly used in intensive care units, may also be important contributors to the development of invasive yeast infections (IYI) in COVID-19 patients [28].

Comparing the results of our study with another conducted by Bhopalwala et al. [29], we noticed that both female patients had similar risk factors: both had diabetes and had a severe COVID-19 infection, thus both were admitted in ICU and mechanically ventilated. Our patient developed invasive pulmonary aspergillosis after 1 month in the ICU, and the other patient 8 months later. Both patients survived even if there were differences between the two since our patient received treatment with voriconazole, which has a major therapeutic drug improvement compared to itraconazole [30].

Another study conducted by Machado et al. [31] showed that CAPA develops in ICU patients with COVID-19-related ARDS (acute respiratory distress syndrome) which makes it a considerable risk factor. Also, most of these patients had bacterial superinfections for which they received antibacterial and cortisone treatment prior to the infection with aspergillus fumigatus. Only five out of eight patients received antifungal treatment and four of them received isavuconazole. Among the patients listed in the study, only one had diabetes, hypertension, and obesity, overlapping with our patient’s profile, but unfortunately, all patients from the study had an unfavorable evolution and passed away.

In a study assisted by Seidel et al. [32], the authors presented that, during the second COVID-19 wave in India, the unchecked use of corticosteroids to treat even the mildest COVID-19 symptoms significantly accelerated the spread of mucormycosis and additionally increasing the risk of fungal infections.

In a study conducted in Romania [33] showing the risk of death in hospitalized COVID-19, the PLR (platelet/lymphocyte ratio) and SII (systemic immune-inflammation index) levels were significantly higher in non-surviving patients in the subgroups of obesity and diabetes.

So far, only rehabilitation has been discovered to be potentially useful in reducing the symptoms of long-term COVID-19; however, further research is still needed to confirm the efficacy of any prospective pharmaceuticals derived from ME/CFS (myalgic encephalomyelitis/chronic fatigue syndrome), POTS (postural tachycardia syndrome), and MCAS (mast cell activation syndrome) [34].

Post-COVID-19 POTS may be linked to several pathophysiological processes. They do not conflict with one another, and none of them have been supported by any published evidence. SARS-CoV-2 may infect and kill postganglionic extracardiac neurons, thereby boosting the cardiac sympathetic nervous system’s (SNS) output in a manner similar to neuropathic POTS. Splanchnic venous pooling or the failure of a reflexive mesenteric vasoconstriction during orthostasis are two examples of this [35].

Theoretically, patients with long-standing diabetes, which can result in debilitation on their own, may be predisposed to long COVID-19. Microvascular injury, which is specifically caused by long-term uncontrolled diabetes, can worsen in people with a SARS-CoV-2 infection. Additionally, diabetes increases the risk of developing a serious and critical COVID-19 condition, increases the likelihood of hospitalization, and raises the chances of needing mechanical ventilation support—all of which can increase the risk of long COVID-19 [36]. The presented patient was diagnosed with diabetes 5 years before admission, her last diabetology follow-up was 3 years ago when she had glycosylated hemoglobin at a value of 6.1%. Furthermore, we can assume that the orthostatic tachycardia the patient presented every time mobilization was attempted was also within diabetes or maybe even caused by post-COVID-19 syndrome. It is known that diabetes can cause autonomic neuropathy and a viral prodrome is a well-documented cause of orthostatic tachycardia with 28–41% of patients reporting this [37,38], and it has also been linked to many different infections [39]. In any case, medical professionals need to be aware that orthostatic tachycardia could be a late complication of COVID-19, and that it is crucial to make the correct diagnosis so that these patients can receive the appropriate care. 

Opportunistic fungal infections are a major problem associated with viral co-infections, leading to poor prognosis and a higher mortality rate (19–40% of patients with invasive candidiasis) [40]. Usually, these fungal infections are accompanied by multidrug-resistant bacteria, especially among patients with ARDS and during an ICU admission [41]. In COVID-19 patients, candidemia is a secondary infection that usually results in death. An extremely high all-cause mortality rate (72.5 versus 26.9%) was observed in COVID-19 ICU patients with candidemia compared with non-candidemic controls matched with CAC (COVID-19-associated candidemia) cases according to length of hospitalization prior to candidemia occurrence in a case-control study conducted during the first wave of the pandemic [42]. Another, often life-threatening, yeast infection associated with SARS-CoV-2 is mucormycosis, with one case reported in Romania [43], and in India with 4252 deaths out of 45,431 cases reported in a study conducted by Kumar et al. [44].

Estimates of CAC prevalence for intubated patients are approximately 5% and estimates of CAPA prevalence for COVID-19 ICU patients range from 0 to 33% [45].

## 4. Conclusions

Diabetes, being overweight, immunosuppression, intensive care unit admission with mechanical ventilation, glucocorticoid treatment, and a severe form of COVID-19 represent important risk factors for immunosuppression. The presence of filamentous fungi on a mycology report is highly suggestive of an Aspergillus infection, and the diagnosis can later be confirmed after thoracic surgery by pathological reports. Invasive pulmonary aspergillosis is life-threatening, but even so, our patient managed to survive. 

We hope that through this article we will alert doctors who will encounter this type of infection to quickly test for aspergillosis so that the potentially fatal complications can be avoided.

## Figures and Tables

**Figure 1 reports-07-00025-f001:**
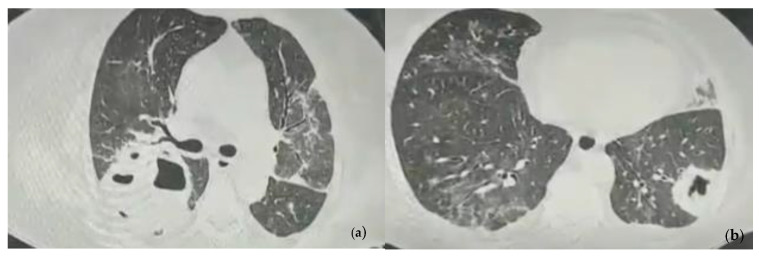
Chest CT images showing pulmonary consolidations with central cavity with heterogeneous air content and polypoid tissue areas at the level of the left and right pulmonary lobes at the admission. (**a**) Consolidation with heterogeneous air content in the right upper lobe, bilateral ground glass infiltration, drainage bronchi, adjacent bronchiectasis; (**b**) Consolidation in the left lower lobe with heterogeneous air content and polypoid tissue accompanied by ground glass infiltration in the contralateral lung.

**Figure 2 reports-07-00025-f002:**
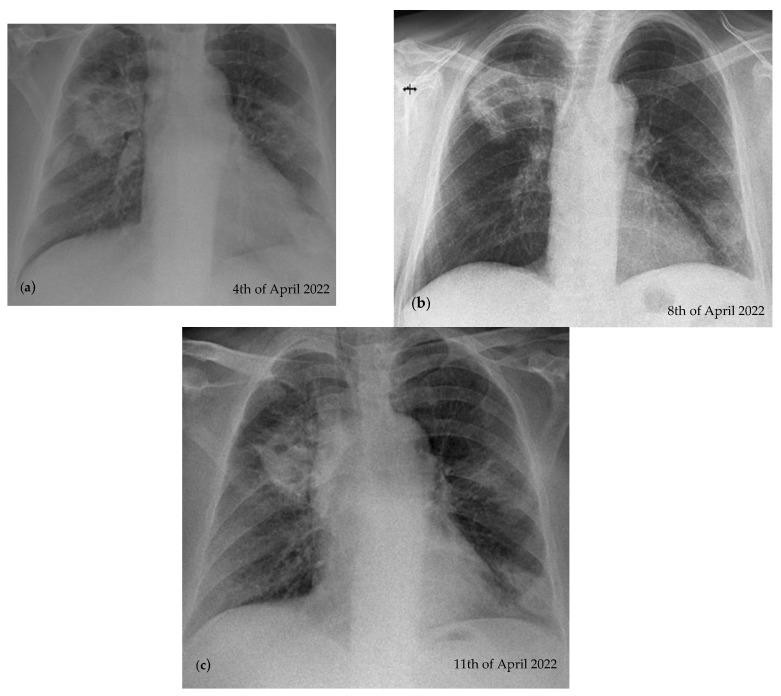
Dynamic reduction in the size of the opacities taken 3 days apart during hospitalization. (**a**) Nodular opacity, irregular, inhomogeneous situated in the 1/3 superior right pulmonary field and in the 1/3 inferior left pulmonary field, cardiomegaly; (**b**) same lesions described in image (**a**), but with a reduction in their size; (**c**) same lesions previously described in image (**b**), but with a reduction in their size.

**Figure 3 reports-07-00025-f003:**
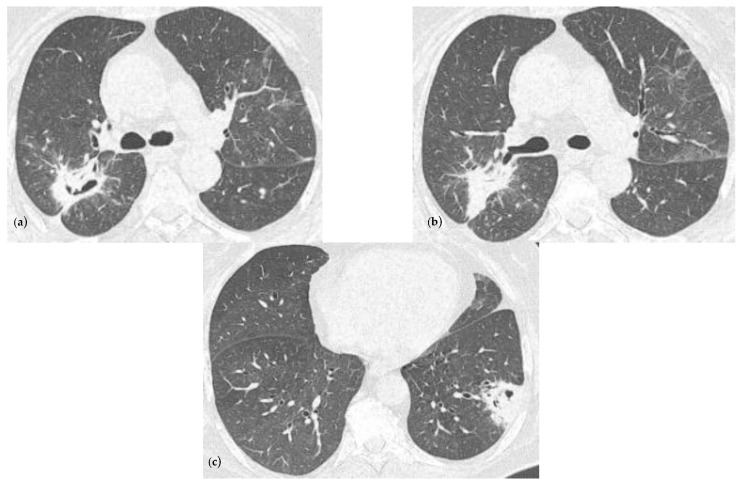
Chest CT imaging at one month after discharge showing a radiological improvement compared to the ones at the admission. (**a**) Consolidation with heterogeneous air content and polypoid tissue in the right upper lobe, but with a reduction in size comparing with Figure 1a, minimal ground glass infiltration in the contralateral lobe; (**b**) same consolidation described in (**a**) but viewing from another slide; (**c**) consolidation in the left lower lobe with heterogenous air content and polypoid tissue, but with a reduction in size comparing with Figure 1b.

**Figure 4 reports-07-00025-f004:**
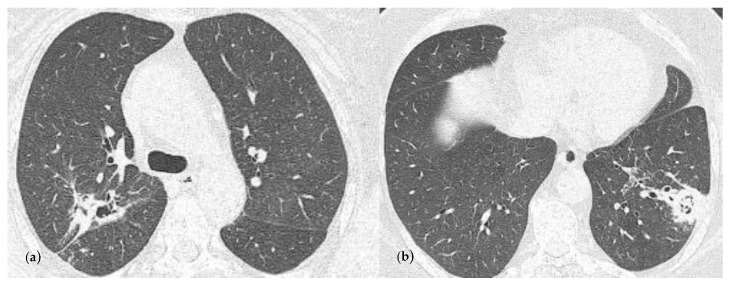
Chest CT imaging at 3 months follow-up and at the end of the antifungal oral treatment. (**a**) Consolidation with heterogeneous air content and polypoid tissue in the right upper lobe, but with a size reduction compared with Figure 3a; (**b**) consolidation in the left lower lobe with heterogenous air content and polypoid tissue, but with a reduction in size comparing with Figure 3c.

**Figure 5 reports-07-00025-f005:**
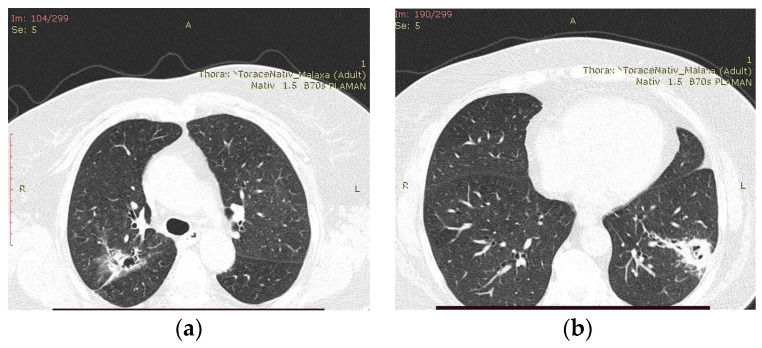
Chest CT examination after 7 months of treatment with voriconazole. (**a**) Fibrocavitary lesion was present in the right upper lobe with the appearance of resolution; (**b**) macronodule with alveolar component, located peripherally in the left lower lobe, with an evolutionary aspect through the extension of peribronchiectatic and reticulomicronodular infiltrates.

**Figure 6 reports-07-00025-f006:**
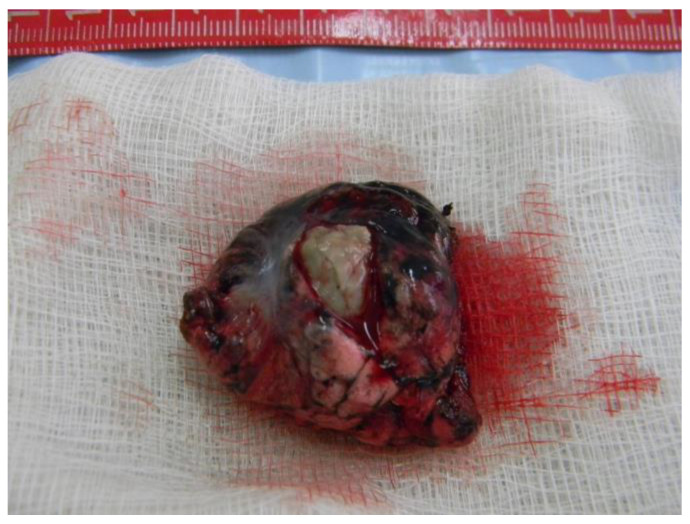
Lung fragment containing Aspergilloma.

**Figure 7 reports-07-00025-f007:**
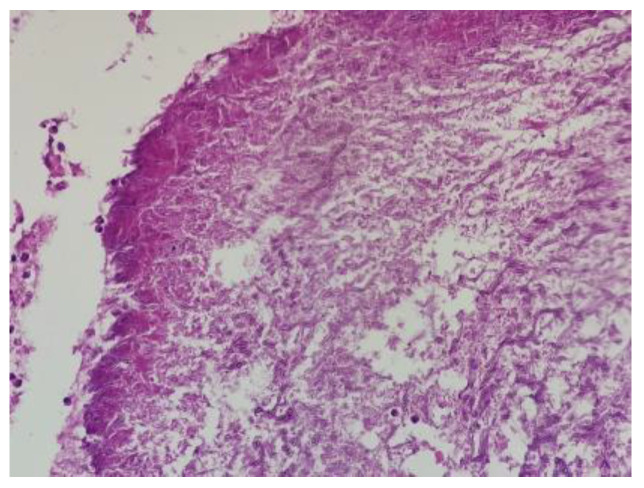
Dichotomous branching (45-degree angle), septate hyphae of Aspergillus; HE; 200×.

**Figure 8 reports-07-00025-f008:**
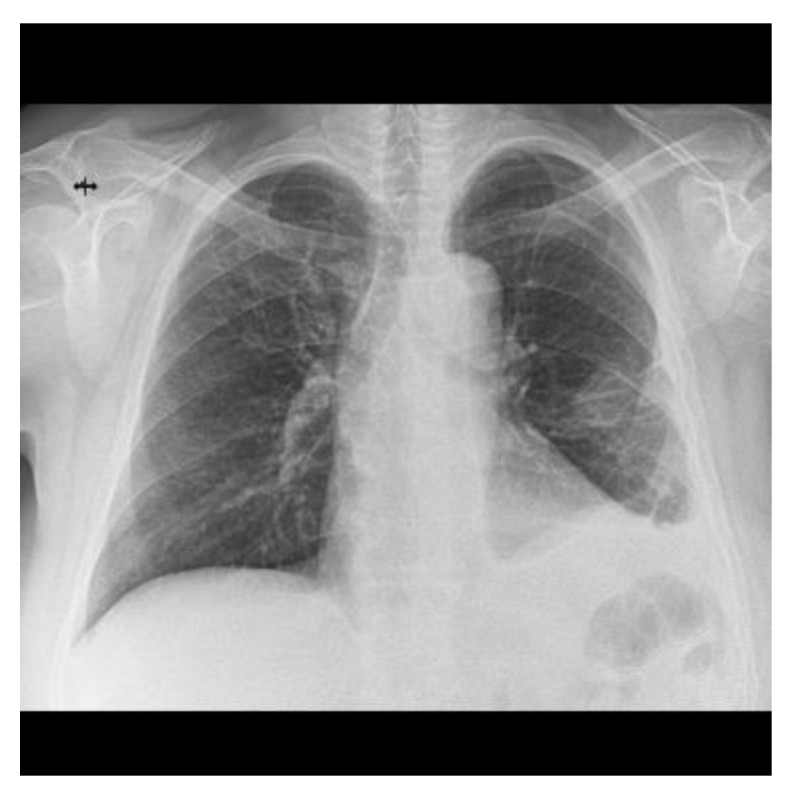
Chest X-ray at the last follow-up compared to serial examinations from Figure 2.

**Table 1 reports-07-00025-t001:** Blood workup with values and normal ranges.

	Values	Normal Range
Blood Glucose	233 md/dL	74–106 mg/dL
Alanine Aminotransferase	55 U/L	0–35 U/L
Aspartate Aminotrasferase	95 U/L	0–35 U/L
Potassium	3.07 mmoL/L	3.5–5.5 mmol/L
Hemoglobin	9.7 g/dL	11.9–14.6 g/dL
D-Dimers	671 ng/mL	0–243 ng/mL
Erythrocyte Sedimentation Rate	41 mm/h	2–30 mm/h

**Table 2 reports-07-00025-t002:** Spirometry with the alveolar-capillary diffusion gas transfer factor.

	Vital Capacity	Gas Transfer Factor
Values	73.6%	20.1%

**Table 3 reports-07-00025-t003:** Administered therapy with doses.

Drug	Dose	Administration
Gentamicin	40 mg/12 h	Intravenous
Tigecycline	50 mg/12 h	Intravenous
Gentamicin + Ipratropium + Physiological Serum	20 mg + 0.5 mg + 1 mL	Aerosols
Voriconazole	200 mg/12 h	Oral
Fragmin	5000 UI/12 h	Subcutaneous
Acetylcysteine	200 mg/8 h	Oral
Seebri Breezehaler	44 mcg/12 h	Inhaler
Metoprolol	50 mg/12 h	Oral
Ivabradine	5 mg/12 h	Oral
Humulin R	Depending on glycemic values	Subcutaneous
Physiological Serum	500 mL/12 h	Intravenous
Glucose 5%	500 mL/12 h	Intravenous
Hepiflor	1 tablet/12 h	Oral

**Table 4 reports-07-00025-t004:** Summarized follow-up findings at 1 month, 3 months, 7 months, and a year after initial discharge.

	1 Month	3 Months	7 Months	1 Year
Blood Glucose	153 mg/dL	130 mg/dL	Not performed	156 mg/dL
Hemoglobin	10.3 g/dL	11.7 g/dL	Not performed	11.7 g/dL
Erytrocyte Sedimentation Rate	108 mm/h	57 mm/h	Not performed	56 mm/h
Spirometry	Normal values	Normal values	Normal values	Normal values
Gas Transfer Factor	53%	67.9%	Not performed	60%
6 Mwt (6 Minute Walk Test)	82.8% of predicted + moderate dyspnea at the end of the test	98.38% + slight fatigue at the end of the test	Not performed	Not performed
Treatment	Voriconazole	Voriconazole	Thoracic surgery	No treatment

**Table 5 reports-07-00025-t005:** Study characteristics of 4 cohort studies and one case report. CAPA = COVID-19-associated pulmonary aspergillosis.

Study Name	Study Type	Study Duration, Country	Description	Steroid Treatment	Capa Treatment	Conclusions
1. Horiuchi H. et al. [9]	Case report	Not mentioned	49-year-old woman, diabetes mellitus, hypertension, hyperuricemia, obesity (BMI 33.8)	methylprednisolone (mPSL) puls	Itraconazole	Patient was transferred to a long-term care facility
2. Permpalung N et al. [10]	Retrospective cohort	March 2020–August 2020, Maryland	39 patients (17 female patients), mean age 66 (55–70), 15 patients had diabetes mellitus, 29 patients had hypertension	26/39; the CAPA patients and the controls statistically significantly differed from one another for hydrocortisone alone (38.5% vs. 12%, respectively; *p* < 0.001).	19/39	17 patients survived
3. Gangneux JP et al. [11]	Retrospective, prospective, observational multicentre cohort.	February–July 2020, France	76 patients (14 female patients), 26 had diabetes, 36 had hypertension, mean age 76; 76/128 had CAPA.	34/76 received steroids treatment.	Voriconazole was given to 44 (76%) patients, liposomal amphotericin B to 20, caspofungin to 16, isavuconazole to 11 (19%), fluconazole to 30 (52%), and other antifungal medications (not specified) to 5 (9%) patients, either separately or in combination.	In addition, 45.8% (95% CI 25.6–67.2) was noted in the 24 patients who had a suitable clinical setting of aspergillosis and non-bronchoalveolar lavage or bronchial or tracheal aspiration positive for *Aspergillus* spp.
4. Xu J et al. [12]	Retrospective cohort	December 2019–April 2020, China	78 patients (29 female patients), mean age 64.3 ± 13.6, 16 had diabetes, 38 had hypertension.	Compared to patients without CAPA, those with CAPA had a significantly higher likelihood of using a higher daily dose (≥40 mg) of methylprednisolone (53.9 vs. 34.2%, *p* = 0.002).	considerably more likely than individuals without CAPA to use methylprednisolone at a higher daily dose (≥40 mg).	37 survived; those with CAPA had a greater ICU death rate (52.6 vs. 28.4%, *p* < 0.001) than those without CAPA.
5. Ruiz-Ruigómez M et al. [13]	Retrospective, observational, single-centre cohort	January 2020–January 2021, Spain	12 patients (3 female patients), mean age 65 ± 10.	All patients received steroid treatment.	Voriconazole was given to 5/12, liposomal amphotericin B to 1/12, and isavuconazol to 2/12 (42.1%). The serum galactomannan test was run in 66.7% (18/27), and 11.1% (2/18) of the results were positive.	It was found that five of the twelve survivors (41.6%) were ultimately diagnosed as colonization, and that either no antifungal treatment was administered or it was stopped in less than 48 h. Ultimately, it was decided that the remaining seven individuals were CAPA-confirmed cases. Upon follow-up, all seven patients were deemed cured and did not exhibit a clinical relapse.

## Data Availability

All data generated or analyzed during this study are included in this published article.

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
