# Peer review of "Wedge Resection and Optimal Solutions for Invasive Pulmonary Fungal Infection and Long COVID Syndrome—A Case Report and Brief Literature Review"

_reports, 2024, doi:10.3390/reports7020025_

Round 1

Reviewer 1 Report

Comments and Suggestions for Authors

The authors presented an interesting case in a patients survived throught critical COVID-19. However, I have several concerns

1. The case report is too lengthy and the description needs to be shorten.

2. Please showed the findings of literature search in the discussion.

3. Please discuss the association between the medicaiton used for critical COVID-19 and the risk of IPA in this case.

Reviewer 2 Report

Comments and Suggestions for Authors

This paper reported a rare case of pulmonary fungal infection in a COVID-19 patient with high risks.

Overall the case presentation was done well, the clinical features, diagnosis and treatment was described in detail. Authors then followed a mini literature review, compared this case with other published cases. Consider of the low number of similar published cases, I suggest that authors could include more paper with not only COVID infection, but also other virus infection in the comparison.

Also the conclusion section uses several long sentences, which are very confusing to readers. Please rephrase and make it concise and clear.

Reviewer 3 Report

Comments and Suggestions for Authors

The present case is well-described, however, the diagnosis of long COVID is unclear. In addition, the authors did a literature search but no structured result was described and summarized. Thus, I cannot found the result of literature search. Lastly, a new table to summarize the findings of the literature serache is needed.

Round 2

Reviewer 1 Report

Comments and Suggestions for Authors

Thank you for the revision.  I have no more query.

Reviewer 3 Report

Comments and Suggestions for Authors

The authors address all my concerns well.